# LINEAR ATTENTION VIA ORTHOGONAL MEMORY

## ABSTRACT

Efficient attentions have greatly improved the computational efficiency of Transformers. However, most existing linear attention mechanisms suffer from an *efficiency degradation* problem, leading to inefficiencies in causal language modeling and hindering their application in long-range language models. This problem is more pronounced under language modeling with unbounded contexts. In this paper, we propose **L**inear **A**ttention **V**ia **O**rthogonal memory (LAVO) to address these limitations, achieving strong performance while maintaining linear complexity. LAVO employs orthogonal decomposition to compress a context into a fixed-size orthogonal memory while effectively minimizing redundancy within the context. Given that orthogonal memory compresses global information, we further dissect the context to amplify fine-grained local information. Additionally, we embed the relative position encoding into LAVO to improve the extrapolation ability. Experimental results show that LAVO greatly improves the efficiency of the causal language model with the best extrapolation performance and outperforms other efficient baselines. Further, we endeavor to employ LAVO for unbounded language modeling and successfully scale the context length to 128K.

## 1 INTRODUCTION

Efficient attention mechanism that has sub-quadratic complexity successfully extends Transformer to longer sequences. Most previous work has proposed to speed up the bidirectional (noncausal) self attention (Choromanski et al., 2021; Lee-Thorp et al., 2022; Qin et al., 2022; Wang et al., 2020; Zaheer et al., 2020; Zhang et al., 2021). Recently, the unprecedented advances made in pretrained large-scale (causal) language models (Brown et al., 2020; Chowdhery et al., 2022; Du et al., 2022; Radford et al., 2018; 2019; Hendrycks et al., 2021; Zhong et al., 2023; An et al., 2023) have drawn considerable attention and stimulated significant interest in the research community. Against this backdrop, there is a growing trend to migrate the focus of linear attention from a noncausal pattern to a causal one to serve as the cornerstone of efficient long-range language models.

In a recent study, however, Zhang et al. (2022) pointed out that very few efficient models meet the demands of autoregressive language modeling. Despite numerous efforts to develop efficient attention mechanisms, only a limited number of available mechanisms focus on modeling causality. Moreover, many existing linear attention mechanisms, such as Long-Short Transformer (Zhu et al., 2021) and cosFormer (Qin et al., 2022), suffer from a problem known as *efficiency degradation*. The problem arises when these efficient attention mechanisms are applied to the case of causal models, leading to a sharp increase in computational complexity, or even reverting back to quadratic complexity (§2). Besides, it gets further exacerbated when given the unbounded context. As such, there remains a bottleneck in the development of more efficient models capable of handling longer contexts.

To address these problems, we propose the l̲inear a̲ttention v̲ia o̲rthogonal memory (LAVO), which achieves strong performance while maintaining linear complexity. We first introduce the orthogonal decomposition to efficiently compress context into a fixed-size orthogonal memory, which maximizes distinguishability among bounded memory units. Considering that orthogonal memory collects coarse-grained global information, we introduce context dissecting to further incorporate the fine-grained local context. In addition, LAVO is equipped with embedded position encoding to obtain good extrapolation capabilities.

We carry out exhaustive experiments to evaluate LAVO covering natural language processing, speech, computer vision, and time-series forecasting. Experiments on language models show that LAVO

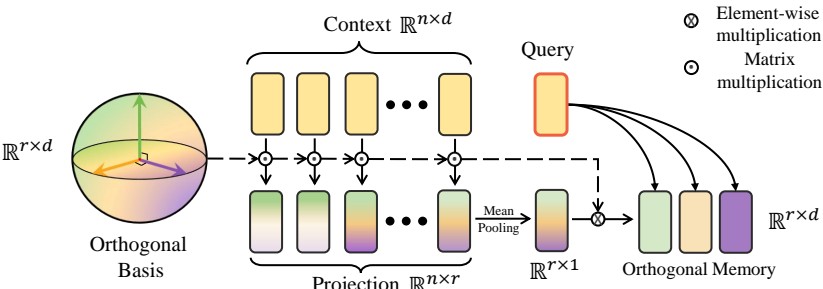

Figure 1: LAVO employs orthogonal decomposition to compress the entire context into a fixed-size memory, referred to as orthogonal memory. A query attends to the orthogonal memory to obtain global information.

outperforms other linear attention on both efficacy and efficiency and achieves good extrapolation ability. Moreover, we evaluate the model as self attention on text-to-speech and summarization tasks. LAVO achieves the best performance on causal text-to-speech and noncausal summarization, and has competitive results on noncausal text-to-speech. Not only self attention, LAVO can also be applied to cross attention. We also conduct the experiments under cross attention pattern on point cloud completion and time-series forecasting tasks, where LAVO outperforms all other attentions. Further, we consider an almost unbounded language modeling task. The empirical results show that LAVO is the only one that can complete the task without IO optimization. All the related codes will be released at `https://github.com/Anonymous`.

## 2 EFFICIENCY DEGRADATION OF CAUSAL LINEAR ATTENTION

### 2.1 BACKGROUND: NONCAUSAL AND CAUSAL ATTENTION

In sequence modeling, noncausal attention has access to the full context. Given the context length $n$, query $\mathbf{Q} = [\mathbf{q}_1, \ldots, \mathbf{q}_n]^\top \in \mathbb{R}^{n \times d}$, key $\mathbf{K} = [\mathbf{k}_1, \ldots, \mathbf{k}_n]^\top \in \mathbb{R}^{n \times d}$, and value $\mathbf{V} = [\mathbf{v}_1, \ldots, \mathbf{v}_n]^\top \in \mathbb{R}^{n \times d}$, vanilla attention (Vaswani et al., 2017) calculates the attention score of the $t$-th query as follows:

$$\text{Attn}(\mathbf{q}_t, \mathbf{K}, \mathbf{V}) = \text{softmax}(\mathbf{q}_t^\top \mathbf{K}^\top)\mathbf{V} \tag{1}$$

Many efficient attention mechanisms are proposed to encode the whole context to speed up vanilla attention, such as Performer (Choromanski et al., 2020) and LARA (Zheng et al., 2022b). On the other hand, modeling causality denotes that the current query vector can only observe previous tokens, which is widely used in language models. Specifically, the attention score of $\mathbf{q}_t$ depends on keys $\mathbf{K}_{\leq t} = [\mathbf{k}_1, \mathbf{k}_2, \ldots, \mathbf{k}_t]^\top$ and values $\mathbf{V}_{\leq \mathbf{t}} = [\mathbf{v}_1, \mathbf{v}_2, \ldots, \mathbf{v}_t]^\top$ before time $t$ as follows:

$$\text{Attn}(\mathbf{q}_t, \mathbf{K}_{\leq t}, \mathbf{V}_{\leq t}) = \text{softmax}(\mathbf{q}_t^\top \mathbf{K}_{\leq t}^\top)\mathbf{V}_{\leq t} \tag{2}$$

Due to causal constraints, these efficient attention mechanisms are required to re-encode the context exclusively for each query and thus lead to significant memory and computation wastage, rendering causal efficient attention less efficient compared to their noncausal counterparts.

### 2.2 MOTIVATION: EFFICIENCY DEGRADATION PROBLEM

Previously, various linear attention mechanisms (Ali et al., 2021; Beltagy et al., 2020; Choromanski et al., 2020; Qin et al., 2022; Wang et al., 2020; Xiong et al., 2021; Zaheer et al., 2020; Zheng et al., 2023) demonstrated their efficiency in long-range modeling. However, these efficiency discussions mostly focus on noncausal or nonautoregressive pattern which encodes sequences as a whole. In contrast, large-scale autoregressive language models such as GPT (Radford et al., 2018) perform attention only on historical texts for generation purposes. A recent study (Zhang et al., 2022) shows that only a few linear attention can perform causal attention, and causal linear attentions such as cosFormer (Qin et al., 2022) and RFA (Choromanski et al., 2020; Peng et al., 2021) would be inefficient in autoregressive language modeling due to large constant. Additionally, some linear

noncausal attention degenerates to $\mathcal{O}(n^2)$ since the causal attention requires recurrent computation. For example, Long-Short Transformer (Zhu et al., 2021) obtains the low-rank matrices from the whole context in noncausal attention, having the complexity of $\mathcal{O}(n)$. In causal attention, it divides the context into multiple segments with a fixed length $l$ and obtains the low-rank matrices for each segment, resulting in the theoretical complexity of $\mathcal{O}(n^2/l)$. The detailed analysis can be found in Appendix A.

Furthermore, consider a more challenging task of causal attention with *unbounded* context, which implies an underlying realistic task of long-range language modeling with extrapolation. Unbounded language models are expected to scale the sequence length to thousands or even more times the upper limit of the current language models (Li et al., 2023), which greatly increases the amount of information in the input context processed by the model. We give the task statement of the unbounded language modeling task as follows.

**Task Statement**  The unbounded language modeling task aims to develop a model that can predict the next token given an unlimited or unbounded amount of preceding context. Formally, given an unbounded context $\mathbf{X} = [\mathbf{x}_1, \mathbf{x}_2, \ldots, \mathbf{x}_n]^\top$ where $n \to \infty$ represents the length of the context, an unbounded language model predicts the next token $\mathbf{y}_{next}$ based on probability $p(\mathbf{y}_{next}|\mathbf{X})$.

When facing the challenge of unbounded language modeling, even certain advanced linear attention such as EVA (Zheng et al., 2023) also degenerates to $\mathcal{O}(n^2)$ complexity. EVA has the complexity of $\mathcal{O}(nc)$ in noncausal attention, where $c$ denotes the number of blocks from the historical context. In the unbound language modeling, the block size cannot be linearly related to $n$. Thus the block size is constant, and $c$ is linearly related to $n$. Finally, EVA degenerates to the complexity of $\mathcal{O}(n^2)$.

Our work is driven by the primary objective of resolving the efficiency degradation problem that arises when linear attention mechanisms are employed in language modeling. Additionally, we elucidate a potential method for surmounting the challenge of unbounded language modeling.

# 3 Linear Attention via Orthogonal Memory

This section is organized as follows: §3.1 describes the process of compressing context into an orthogonal memory and proposes LAVO with linear complexity in self attention. Then we introduce the context dissecting (§3.2) and embedded position encoding (§3.3) to enhance LAVO. Finally, we present how LAVO performs as cross attention in §3.4.

## 3.1 Context Compression via Orthogonal Decomposition

The attention mechanism enables each token to retrieve relevant information from the context memory. Given a context $\mathbf{X} \in \mathbb{R}^{n \times d}$ with length $n$, vanilla attention (Vaswani et al., 2017) first obtains queries $\mathbf{Q} \in \mathbb{R}^{n \times d}$, keys $\mathbf{K} \in \mathbb{R}^{n \times d}$, and values $\mathbf{V} \in \mathbb{R}^{n \times d}$ by $\mathbf{X}W_q = [\mathbf{q}_1, \ldots, \mathbf{q}_n]^\top$, $\mathbf{X}W_k = [\mathbf{k}_1, \ldots, \mathbf{k}_n]^\top$ and $\mathbf{X}W_v = [\mathbf{v}_1, \ldots, \mathbf{v}_n]^\top$, respectively. Then a query $\mathbf{q}_t$ attends to $\mathbf{K}$ and $\mathbf{V}$ as Eq. 1. The vanilla attention has a quadratic complexity of $\mathcal{O}(n^2)$. One widely-used method to improve efficiency is to crop or compress the context into a fixed-size memory (Luong et al., 2015a; Peng et al., 2022a; Sukhbaatar et al., 2021b; Wang et al., 2021; 2020), which limits the amount of context retrieved by each query. In this way, the distinguishability of the vectors in the bounded memory determines the richness of stored information.

We use context compression via orthogonal decomposition (CODE) to build a distinguishable bounded memory, improving the information entropy of compressed context. We first divide the bounded memory into several orthogonal spaces and then project the token features into these spaces to obtain the memory vectors. The orthogonal spaces maximize the feature distinguishability in the bounded memory. Specifically, we initialize a set of orthogonal bases $\mathbf{B} = [\mathbf{b}_1, \ldots, \mathbf{b}_r]^\top \in \mathbb{R}^{r \times d}$ as introduced in (Saxe et al., 2013), where $r < n$ denotes the numbers of orthogonal bases. Then we compress the context $\mathbf{X} = [\mathbf{x}_1, \ldots, \mathbf{x}_n]^\top \in \mathbb{R}^{n \times d}$ as follows:

$$\widetilde{\mathbf{X}} = \mathrm{CODE}(\mathbf{X}) = \mathbf{B} \odot \mathbf{H} \in \mathbb{R}^{r \times d}, \quad \mathbf{H} = \frac{1}{n} \sum_{t=1}^{n} \mathbf{B} \cdot \mathbf{x}_t \in \mathbb{R}^{r \times 1} \tag{3}$$

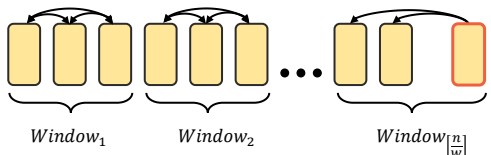

Figure 2: Context dissection in LAVO. The context is divided into multiple windows of size $w$.

where $\odot$ denotes element-wise multiplication and $\widetilde{\mathbf{X}}$ indicates the orthogonal memory compressed from context. Memory vectors are mutually orthogonal in LAVO, not in ABC (Peng et al., 2022a). The orthogonal bases can be seen as distributed representation spaces. Each token feature is decomposed into each orthogonal space. The averaged projection $\mathbf{H}$ reflects the information from the context contained in each orthogonal space. Finally, query $\mathbf{q}_t$ attends to orthogonal memory to obtain global information as $\text{LAVO}(\mathbf{q}_t, \mathbf{X}) = \text{softmax}(\mathbf{q}_t^\top \widetilde{\mathbf{X}}^\top)\widetilde{\mathbf{X}}$. Further, we can produce the causal attention form of LAVO as follows:

$$\text{LAVO}(\mathbf{q}_t, \mathbf{X}_{\leq t}) = \text{softmax}(\mathbf{q}_t^\top \widetilde{\mathbf{X}}_{\leq \mathbf{t}}^\top)\widetilde{\mathbf{X}}_{\leq \mathbf{t}}, \ \ \widetilde{\mathbf{X}}_{\leq t} = \mathbf{B} \odot \mathbf{H}_t, \ \ \mathbf{H}_t = \frac{(t-1)\mathbf{H}_{\mathbf{t-1}} + \mathbf{B} \cdot \mathbf{x}_t}{t} \quad (4)$$

In causal attention, LAVO has the complexity of $\mathcal{O}(rn)$ as causal attention, where $r$ is a constant.

## 3.2 ENHANCING LAVO WITH CONTEXT DISSECTION

Previous work (Zhang et al., 2022) shows that local attention plays an important role in improving the performance of the model. Although orthogonal memory covers the information of the whole context, it cannot produce fine-grained local content. Therefore, we combine orthogonal memory and local context. As shown in Figure 2, we first dissect the context to multiple windows $[\mathbf{C}_{1:w}, \mathbf{C}_{w+1:2w}, \dots, \mathbf{C}_{\lfloor n/w \rfloor w:(\lfloor n/w \rfloor + 1)w}]$ with size $w$ and divide context into local context and global context for a query $\mathbf{q}_t$. In causal attention, $\mathbf{C}_{\lfloor t/w \rfloor w:t}$ forms a local context, and $[\mathbf{C}_{1:w}, \dots, \mathbf{C}_{(\lfloor t/w \rfloor - 1)w:\lfloor t/w \rfloor w}]$ forms a global context. Then $\mathbf{q}_t$ attends to $\mathbf{C}_{\lfloor t/w \rfloor w:t}$ to obtain local features $\mathbf{F}_{local}$ as $\text{Attn}(\mathbf{q}_t, \mathbf{C}_{\lfloor t/w \rfloor w:t} W_k, \mathbf{C}_{\lfloor t/w \rfloor w:t} W_v)$, which has the time complexities of $\mathcal{O}(wn)$. With the independent window partition, the first token in the window cannot attend the previous context and the last token cannot attend the subsequent context. Thus, we extend the $w$ tokens before the first token in the window and use a mask to ensure that each token attends to $w$ tokens.

Due to the substantial variability observed among individual tokens, the compressed orthogonal memory is likewise characterized by a significant degree of variance. We perform local attention on the tokens in each window before compressing the global context to aggregate window information and reduce the variance. Then we use LAVO to obtain the global feature $\mathbf{F}_{global}$ from them. Finally, we average $\mathbf{F}_{local}$ and $\mathbf{F}_{global}$ as the outputs. The total time complexity is $\mathcal{O}((w+r)n)$.

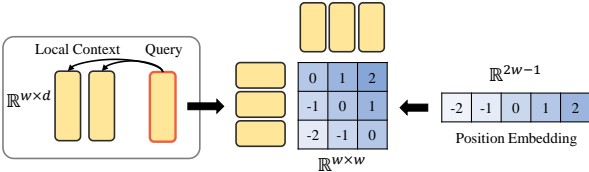

Figure 3: Embedded position encoding in LAVO. Position encoding models the relative positional relationship between tokens in a window.

## 3.3 EMBEDDED POSITION ENCODING

The extrapolation ability of language models holds paramount importance in their practical applications, especially for unbounded language models. Language models with strong extrapolation ability can generate coherent and contextually appropriate text beyond the confines of their training data, making them more versatile and adaptable to real-world scenarios. Relative position embedding plays a pivotal role in bolstering the extrapolation ability of language models. However, most relative position leads to the time and space complexities of $\mathcal{O}(n^2)$. As discussed above, we embed the position

Table 1: Hyperparameters of tasks and models. num_bases denotes the number of orthogonal bases.

| Task | TTS | LSTF | | PCC | Summ | LM |
|---|---|---|---|---|---|---|
| Data | LJSpeech | ETT | Weather | PCN | Multi-News | PG-19 |
| *Training Hyperparameters* | | | | | | |
| Batch Size | 48 | 32 | | 48 | 64 | 16 |
| Number of Steps | 20K | 6 (epochs) | | 300 (epochs) | 50K | 125K |
| Warmup Steps | 4K | - | | 4K | 1K | 10K |
| Peak Learning Rate | 5e-4 | 1e-4 | | 5e-4 | 5e-4 | 5e-4 |
| Scheduler | Inverse Sqrt | Exponential Decay | | LambdaLR | Inverse Sqrt | Inverse Sqrt |
| Optimizer | AdamW | AdamW | | AdamW | AdamW | AdamW |
| Adam | (0.9,0.98) | (0.9,0.999) | | (0.9,0.98) | (0.9,0.98) | (0.9,0.999) |
| Clip Norm | 5.0 | 0.0 | | 0.0 | 0.0 | 0.0 |
| Attention Dropout | 0.1 | 0.05 | | 0.0 | 0.1 | 0.1 |
| Weighty Decay | 0.01 | 0.0 | | 5e-4 | 0.01 | 0.01 |
| Tokens per Batch | - | - | | - | - | $2^{17}$ |
| Iteration | - | 5 | 3 | - | - | - |
| *backbone-specific Hyperparameters* | | | | | | |
| # Attention heads | 8 | 8 | 8 | 6 | 8 | 12 |
| Hidden size | 512 | 512 | 512 | 768 | 512 | 768 |
| Hidden size in FFN | 2048 | 2048 | 2048 | 3072 | 2048 | 3072 |
| # Encoder Layers | 6 | 2 | 3 | 6 | 6 | - |
| # Decoder Layers | 6 | 1 | 2 | 6 | 6 | 12 |
| *Model-specific Hyperparameters* | | | | | | |
| d_state (S4D) | 64 | 16 | | 64 | 64 | 64 |
| wsize (local, LongShort) | 16 | 16 | | 16 | 16 | 16 |
| landmarks (ABC, LongShort, LARA) | 64 | 16 | | 64 | 64 | 64 |
| attn_dim (Performer) | 64 | 16 | | 64 | 64 | 64 |
| num_bases (LAVO) | 64 | 16 | | 64 | 64 | 64 |

encoding into local attention of LAVO. Specifically, we first initialize relative position embeddings $\mathbf{P} \in \mathbb{R}^{2w-1}$. According to the distance between $\mathbf{q}_i$ and $\mathbf{k}_j$ in a window, we add $\mathbf{P}_{j-i+w-1}$ to the attention score $\mathbf{A}_{i,j}$, as shown in Figure 3. Therefore, the time and space complexities of embedded position encoding are $\mathcal{O}(wn)$ and $\mathcal{O}(w)$, respectively.

## 3.4 EXTENSION TO CROSS ATTENTION

Since query and key of cross attention are derived from target sequence $\mathbf{Y} \in \mathbb{R}^{m \times d}$ and source sequence $\mathbf{X} \in \mathbb{R}^{n \times d}$, respectively, there is no potential alignment information between queries and keys. It is also challenging to apply linear attention to cross attention. With context compression via orthogonal decomposition, LAVO only compresses the source sequence and can be easily adapted to cross attention. Given orthogonal bases $\mathbf{B} \in \mathbb{R}^{r \times d}$, LAVO can be extended to the setting of cross attention as follows:

$$\mathbf{Q} = W_q \mathbf{Y}, \quad \widetilde{\mathbf{X}} = \text{CODE}(\mathbf{X}) \tag{5}$$

$$\text{LAVO}(\mathbf{Q}, \mathbf{X}) = \text{softmax}(\mathbf{Q}\widetilde{\mathbf{X}}^\top)\widetilde{\mathbf{X}} \tag{6}$$

where $W_q \in \mathbb{R}^{d \times d}$ is a learnable parameter. Note that in this case, local features in LAVO are removed since there is no potential alignment information. The complexity of compressing the source sequence and calculating the attention is $\mathcal{O}(rm)$ and $\mathcal{O}(rn)$, respectively. The total complexity of LAVO as cross attention is $\mathcal{O}(r(n+m))$, where $r$ is a constant.

## 4 EXPERIMENTS

We conduct extensive experiments covering natural language processing, speech, computer vision, and time-series forecasting. We first conduct the experiments under self attention pattern, including language modeling, text-to-speech, and summarization tasks. Then we evaluate the performance of the proposed method under the cross attention pattern. Finally, we conduct an experiment on the language modeling task with an extremely long context and analyze the impact of each component on the model performance. We compare LAVO with ten strong baseline models, including FlashAttention (Dao et al., 2022), local attention (Luong et al., 2015a), LongShort (Zhu et al., 2021), S4D (Gu et al., 2022c), ProbSparse (Zhou et al., 2021), Performer (Choromanski et al., 2020), cosFormer (Qin et al.,

Table 2: Language modeling perplexity on the PG-19 dataset. During the training phase, a context with a length of 8,192 is input, while during the test phase, models are exposed to contexts with lengths of 12,288 and 16,384 to evaluate their extrapolation ability. The vanilla attention fails on LM tasks due to out-of-memory. Memory cost (Mem.) and speedup (Sp.) are measured with $1\times80$GB A100. FA + RoPE denotes the FlashAttention with rotary position embedding (Su et al., 2021). Bold indicates the best performance, and underline indicates the best performance in linear attention.

| Complexty | Model | #Params | Context Length | | | | | | | | |
|---|---|---|---|---|---|---|---|---|---|---|---|
| | | | 8192 | | | 12288 | | | 16384 | | |
| | | | PPL↓ | Mem. | Sp. | PPL↓ | Mem. | Sp. | PPL↓ | Mem. | Sp. |
| $\mathcal{O}(n^2)$ | vanilla | 122.32M | - | 7107M | 1.0× | - | 14023M | 1.0× | - | 25747M | 1.0× |
| | FlashAttention | 122.32M | 16.02 | **4627M** | **14.0×** | 40.33 | **6847M** | 16.1× | 94.34 | **9133M** | 18.7× |
| | FA + RoPE | 122.32M | **15.09** | 4679M | 12.4× | 20.11 | 6892M | 14.5× | 34.84 | 9193M | 16.96× |
| | LongShort | 136.61M | 15.52 | 5015M | 3.5× | **19.17** | 7809M | 4.0× | 26.06 | 10929M | 5.7× |
| $\mathcal{O}(n\log n)$ | S4D | 155.41M | 15.78 | 7975M | 1.3× | 51.96 | 11751M | 1.6× | - | 16111M | 2.3× |
| $\mathcal{O}(n)$ | ABC | 122.42M | 31.53 | 6165M | 2.2× | 86.78 | 9373M | 3.1× | 147.43 | 12669M | 4.1× |
| | local | 122.32M | 19.73 | 5673M | 5.0× | 21.24 | 8759M | 6.7× | 22.16 | 11909M | 9.1× |
| | LAVO | 122.91M | 19.43 | 4688M | 11.49× | 19.41 | 6904M | **16.3×** | **19.40** | 9337M | **22.0×** |

2022), LARA (Zheng et al., 2022b), Nyströmformer (Xiong et al., 2021), and ABC (Peng et al., 2022b). We replace the attention modules in backbone models with efficient attentions and use the same experimental settings to verify the performance of models. We follow the setting of (Zhang et al., 2022) to maintain a fair comparison between each efficient model. Details of task setting and baselines can be found in Table 1. We also evaluate LAVO on Long Range Arena (Tay et al., 2020b) in Appendix C.

## 4.1 SELF ATTENTION

**Language Modeling** We carry out the language modeling experiment to evaluate the efficiency and efficacy of causal efficient attention on long context modeling. we select the PG-19 dataset (Rae et al., 2019) which consists of books extracted from the Project Gutenberg books library. The train/valid/test sets contain 28,602/50/100 books, respectively. GPT-2 (Radford et al., 2019) with a 12-layer decoder is used to serve as the backbone model, and token-level perplexity (PPL) is selected to evaluate the efficient attentions. We use a BPE-based GPT-2 tokenizer with a vocabulary size of 50,257. We feed the model a context of length 8,192 during the training phase and use contexts of length 8,192, 12,288, and 16,384 during the testing phase to evaluate the extrapolation ability of the model. We remove the sinusoidal position embedding in the LAVO-based GPT-2 since embedded position encoding has similar capabilities to it. Table 2 shows the results of the language modeling task. Results show that LAVO has the lowest perplexity compared with ABC and local which are also of linear complexity. In addition, the perplexity of LAVO gradually decreases with the increase of sequence length, while other models increase to varying degrees. Notably, although FlashAttention with rotary position embedding (FA + RoPE) also uses the relative position embedding, its perplexity still increases significantly with the increase of the sequence. This suggests that a longer context allows LAVO to improve language modeling capability, showing that LAVO has a strong extrapolation ability. We can also find that LAVO greatly reduces memory cost with significant speedup and achieves speedups of 20.3× and 9,337MB memory cost based on a context of length 16,384, second only to FlashAttention which uses IO optimization.

**Text-to-Speech** We conduct the text-to-speech experiment to assess models under both causal and noncausal self patterns. In this task, we use LJSpeech dataset (Ito, 2017) which has the average sequence length of 559, and apply Transformer-TTS (Li et al., 2019) as the backbone model. Following Zhang et al. (2022), we evaluate the performance of speech synthesis by Mel Cepstral Distortion (MCD) and Mel Spectral Distortion (MSD). We show the results under causal self pattern in Table 3, LAVO with the linear complexity significantly improves the performance of vanilla attention by -0.085 MCD and outperforms the other efficient attentions. Moreover, we replace the self attention in the encoder of Transformer-TTS encoder to conduct noncausal self pattern experiments. The results under noncausal self pattern are shown in Table 5. We can find that LAVO outperforms most previous baselines and achieves comparable results with state-of-the-art LongShort.

Table 3: Automatic evaluation results under causal self pattern on text-to-speech task. The best results are bolded.

| Complexity | Model | #Params | MCD↓ | MSD↓ |
|---|---|---|---|---|
| $\mathcal{O}(n^2)$ | vanilla | 54.40M | 4.095 | 2.199 |
| | FlashAttention | 54.40M | 4.066 | 2.207 |
| | LongShort | 57.57M | 4.039 | 2.195 |
| $\mathcal{O}(n \log n)$ | S4D | 55.20M | 4.030 | 2.189 |
| $\mathcal{O}(n)$ | ABC | 54.50M | 4.058 | 2.189 |
| | local | 54.40M | 4.141 | 2.221 |
| | LAVO | 54.60M | **4.010** | **2.179** |

Table 4: Automatic evaluation results under causal self pattern on summarization task. The best results are bolded. underline indicates the best performance in linear attention. (†) the performance drop compared to vanilla is possibly due to precision errors.

| Complexity | Model | #Params | R-1↑ | R-2↑ | R-L↑ |
|---|---|---|---|---|---|
| $\mathcal{O}(n^2)$ | vanilla | 123.70M | 34.61 | 6.35 | 31.66 |
| | FlashAttention† | 123.70M | 34.25 | 6.24 | 31.32 |
| | LongShort | 126.88M | 33.55 | 6.27 | 30.71 |
| $\mathcal{O}(n \log n)$ | S4D | 131.59M | **34.90** | **6.65** | **31.98** |
| $\mathcal{O}(n)$ | ABC | 123.75M | 30.17 | 5.48 | 27.92 |
| | local | 123.70M | 33.50 | 6.27 | 30.74 |
| | LAVO | 123.89M | 34.16 | 6.16 | 31.16 |

Table 5: Automatic evaluation results under noncausal self pattern on text-to-speech task. The best results are bolded. Underline denotes the second-ranked result.

| Complexity | Model | #Params | MCD↓ | MSD↓ |
|---|---|---|---|---|
| $\mathcal{O}(n^2)$ | vanilla | 54.40M | 4.095 | 2.199 |
| | FlashAttention | 54.40M | 4.077 | 2.175 |
| $\mathcal{O}(n \log n)$ | S4D | 55.20M | 4.017 | 2.195 |
| | ProbSparse | 54.40M | 4.034 | 2.161 |
| $\mathcal{O}(n)$ | LARA | 54.40M | 4.116 | 2.209 |
| | cosFormer | 54.40M | 4.030 | 2.160 |
| | Performer | 54.40M | 4.115 | 2.198 |
| | LongShort | 55.20M | **3.913** | **2.136** |
| | Nyströmformer | 54.40M | 4.274 | 2.276 |
| | local | 54.40M | 4.015 | 2.164 |
| | ABC | 54.50M | 4.085 | 2.204 |
| | LAVO | 54.60M | 3.964 | 2.155 |

Table 6: Automatic evaluation results under noncausal self pattern on summarization task. The best results are bolded.

| Complexity | Model | #Params | R-1↑ | R-2↑ | R-L↑ |
|---|---|---|---|---|---|
| $\mathcal{O}(n^2)$ | vanilla | 123.70M | 34.61 | 6.35 | 31.66 |
| | FlashAttention | 123.70M | 34.64 | 6.52 | 31.66 |
| $\mathcal{O}(n \log n)$ | ProbSparse | 123.70M | 34.62 | 6.36 | 31.64 |
| $\mathcal{O}(n)$ | LARA | 123.70M | 34.03 | 6.23 | 31.23 |
| | cosFormer | 123.70M | 34.77 | 6.34 | 31.74 |
| | Performer | 123.70M | 34.85 | 6.54 | 31.88 |
| | LongShort | 124.11M | 34.35 | 6.41 | 31.55 |
| | Nyströmformer | 123.70M | 34.45 | 6.30 | 31.56 |
| | local | 123.70M | 38.50 | 10.54 | 35.39 |
| | ABC | 123.75M | 33.80 | 6.07 | 30.98 |
| | LAVO | 123.89M | **39.01** | **10.77** | **35.70** |

Table 7: Automatic evaluation results under cross attention pattern on point cloud completion task. The best results are bolded.

| Complexity | Model | CDL1↓ | CDL2↓ | F-score↑ |
|---|---|---|---|---|
| $\mathcal{O}(n^2)$ | vanilla | 7.425 | 0.231 | 0.779 |
| $\mathcal{O}(n)$ | ABC | 7.344 | 0.227 | 0.784 |
| | Performer | 7.368 | 0.235 | 0.783 |
| | cosFormer | 8.673 | 0.298 | 0.704 |
| | LAVO | **7.200** | **0.216** | **0.794** |

Table 8: Automatic evaluation results under cross attention pattern on time-series forecasting task. The best results are bolded.

| Complexity | Model | ETT | | Weather | |
|---|---|---|---|---|---|
| | | MSE↓ | MAE↓ | MSE↓ | MAE↓ |
| $\mathcal{O}(n^2)$ | vanilla | 1.138 | 0.775 | 0.478 | 0.508 |
| $\mathcal{O}(n)$ | ABC | 1.147 | 0.809 | 0.489 | 0.520 |
| | Performer | 1.254 | 0.819 | 0.463 | 0.508 |
| | cosFormer | 1.219 | 0.823 | 0.474 | 0.509 |
| | LAVO | **1.123** | **0.765** | **0.444** | **0.492** |

**Summarization** In order to further evaluate the comprehension ability of long contexts, we carry out experiments on the summarization task under both causal and noncausal self patterns. We select Multi-News datasets (Fabbri et al., 2019) for this task. The maximum context and summary lengths are set to 4,096 and 400, respectively. We use Transformer (Vaswani et al., 2017) with 6-layer encoder and 6-layer decoder as the backbone model and ROUGE (R-N) (Lin, 2004) as the evaluation metric. The results under causal self pattern are shown in Table 4. We can find that FlashAttention has high ROUGE and S4D with the complexity of $\mathcal{O}(n \log n)$ performs better than other attentions. This indicates that it is a great challenge for efficient attention with linear time on the summarization task. However, LAVO achieves the best performance against other linear attentions and has competitive results with FlashAttention. Additionally, we show the results under the noncausal self pattern in Table 6. Results show that LAVO significantly improves the performance of Transformer by +4.4 R-1, +4.42 R-2, and 4.04 R-L, respectively, indicating LAVO has a strong long context encoding capability.

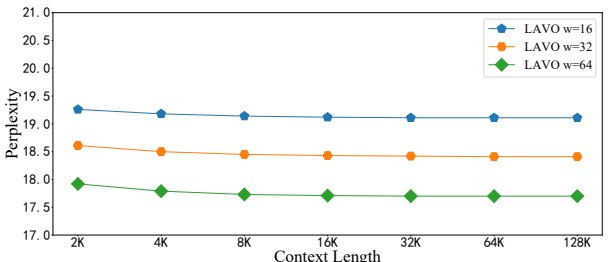

Figure 4: Unbounded language modeling task. We train LAVO with a context length of 2,048. During the test phase, we input a sequence with a maximum length of 128K to calculate perplexity.

Table 9: Ablation study on Summarization task. The best results are bolded. LAVO w/o epe denotes removing the embedded position embedding, while LAVO w/o epe, cd denotes further removing the context dissecting.

| Model | R-1↑ | R-2↑ | R-L↑ |
|---|---|---|---|
| LAVO | **39.19** | **10.82** | **35.95** |
| LAVO w/o epe | 39.01 | 10.77 | 35.70 |
| LAVO w/o epe, cd | 35.29 | 6.80 | 32.29 |

## 4.2 CROSS ATTENTION

Cross attention shows the ability of the model to combine non-homologous information. However, only a few works focus on efficient cross attention. LAVO uses context compression based on orthogonal decomposition and can easily work as cross attention. To evaluate the performance of cross attention, we conduct experiments on two tasks: point cloud completion and time-series forecasting.

**Point Cloud Completion** We use the PCN dataset (Griffiths & Boehm, 2019) for this task. Following (Zhang et al., 2022), we select PoinTr (Yu et al., 2021) as the backbone network, and use Chamfer-Distance (Huang et al., 2020) and F-score (Tatarchenko et al., 2019) as the measurements. We manually downsample the number of input points to 1,536 using a convolution module and set 3,584 point proxies as the decoder input and retain the other settings as (Yu et al., 2021). Results in Table 7 indicate that LAVO substantially improves over other baselines in all metrics. It is observed that LAVO surpasses not only the quadratic vanilla baseline but the strong linear ABC model.

**Time-Series Forecasting** We evaluate the model on Weather and ETT datasets (Zhou et al., 2021), where ETT consists of ETT-h1, ETT-h2, ETT-m1. We select the Informer (Zhou et al., 2021) as the backbone model and use Mean Square Error (MSE) and Mean Absolute Error (MAE) as the evaluation metrics. The input/output lengths in weather and ETT datasets are set to 720/720 and 336/720, respectively. We consider both univariate and multivariate evaluations. To obtain the final score, we average scores on univariate and multivariate settings. For the ETT dataset, we also average the results from the three sub-datasets. We report the results in Table 8. We can find that LAVO outperforms the other attentions on both ETT and weather datasets. Notably, ABC, Performer, and cosFormer perform worse than vanilla on the ETT dataset, while LAVO is the only one whose performance surpasses vanilla.

## 4.3 DISCUSSION

**Unbounded language modeling** We further conduct a challenging task: Unbounded language modeling. Due to GPU memory limitations, we consider almost unbounded language modeling that the model is required to process the extreme length of context on a single 80G A100 GPU. We then feed the model a context of length $\{2K, 4K, \ldots, 128K\}$ to evaluate its performance during the testing phase. We use the same experimental setting and hyperparameters as the language modeling task described in §4.1, except the context length is 2,048 and the batch size is 64. We report the results in Figure 4. We find that LAVO is the only one that can be tested on a context of 128K length without IO optimization. In addition, the results show that the perplexity of LAVO gradually decreased with increasing sequence length, indicating that the language modeling capability of LAVO benefits from increasing context. We also present the perplexity of LAVO with different window sizes. LAVO with a window size of 64 has the lowest perplexity. It suggests that a larger window size tends to lead to better extrapolation capability.

**Ablation Study**  We conduct experiments to evaluate the influence of context dissecting and embedded position encoding on the performance of LAVO. The experimental setting is the same as §4.3. We input the context with a length of 2,048 during the test phase. Table 9 shows the results, where LAVO w/o epe denotes removing the embedded position encoding and LAVO w/o epe, cd denotes removing both embedded position encoding and context dissecting but the local feature is retained. We can find that both context dissecting and embedded position embedding improve the performance of LAVO, and removing context dissecting leads to a larger performance drop.

## 5  RELATED WORK

**Efficient Attention**  Recently, various efficient attention architectures have been proposed to enhance the efficiency of standard attention (Vaswani et al., 2017) due to its quadratic time complexity and memory cost as the sequence length increases. According to different design philosophies, there are sparse attention (Chen et al., 2022; Kitaev et al., 2019; Vyas et al., 2020; Tay et al., 2020a; Roy et al., 2021; Parmar et al., 2018; Xiong et al., 2022; Liu et al., 2021), low-rank attention (Guo et al., 2019; Chen et al., 2020; Xiong et al., 2021; Zheng et al., 2023), recurrence attention (Gu et al., 2022b;a; Zhu et al., 2021; Rae et al., 2020), memory compressison (Peng et al., 2022a; Liu* et al., 2018; Lee et al., 2019; Wang et al., 2020; Ma et al., 2021), kernel-based attention (Choromanski et al., 2020; Katharopoulos et al., 2020; Zheng et al., 2022a; Qin et al., 2022). In addition to reducing the theoretical complexity, some research has accelerated calculations during actual running, such as reducing IO complexity (Dao et al., 2022). However, most previous works have significantly improved the performance of self attention, but only a few works focus on efficient causal attention (Zheng et al., 2023; Zhu et al., 2021; Gu et al., 2022b) and cross-attention (Peng et al., 2022a).

**Attention with Bounded Memory**  The attention mechanism can be viewed as retrieving information from a given context. As the context continues to grow longer, the consumption caused by retrieval is also increasing. One way to obtain the fixed-size context is attending to the nearest $k$ tokens (Beltagy et al., 2020; Sukhbaatar et al., 2021a; Qiu et al., 2020; Luong et al., 2015b). However, they ignore most of the global information. Previous works (Peng et al., 2022a; Wang et al., 2020; Ma et al., 2021) considered compressing unbounded context into a bounded memory. Peng et al. (2022a) and Ma et al. (2021) used a learned weight matrix and a dual attention mechanism to convert the context into a fixed-size memory, and then let queries attend to this memory.

## 6  LIMITATION

Although LAVO has shown promising improvements in causal attention with linear complexity, there are still some problems that remain to be resolved. One of the main challenges is that LAVO still lags a little behind FlashAttention when processing relatively short texts. This means that its efficiency is limited when applied in relatively short sequences. However, this issue can be mitigated by using the same IO optimization methods as FlashAttention, which further achieves a leap in efficiency improvement. We believe that addressing this problem will make LAVO more useful and efficient for sequences with various lengths.

## 7  CONCLUSIONS

In this paper, we discuss the efficiency degradation of causal linear attention, especially in unbounded language models. To address this problem, we propose linear attention via orthogonal memory (LAVO) to achieve strong performance preserving linear complexity. LAVO compresses the context via orthogonal decomposition to produce bounded orthogonal memory with distinguishable features. To further enhance the encoding ability, we introduce context dissecting to incorporate fine-grained local context. Moreover, we embed a position encoding into LAVO to improve the extrapolation ability of causal attention. We conduct various experiments on self and cross attention, where LAVO exhibits strong self-encoding and cross-encoding capabilities. Notably, LAVO outperforms other linear attention and significantly improves the efficiency and extrapolation ability in language modeling. Further, we also consider an experiment on almost unbounded language modeling. Results show that LAVO achieves a language modeling task with a context length of 128K, which is almost impossible for other models, except FlashAttention using IO optimization.

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

## A    EFFICIENCY DEGRADATION

In autoregressive models, causal attention is required, but prior works like RFA and LongShort suffer from efficiency degradation due to recurrent computation. Under noncausal attention pattern, RFA calculates attention as follows:

$$\text{RFA}\left(\boldsymbol{q}_t, \{\boldsymbol{k}_i\}, \{\boldsymbol{v}_i\}\right) = \frac{\phi\left(\boldsymbol{q}_t\right)^\top \sum_i \phi\left(\boldsymbol{k}_i\right) \otimes \boldsymbol{v}_i}{\phi\left(\boldsymbol{q}_t\right) \sum_j \phi\left(\boldsymbol{k}_j\right)} \tag{7}$$

As causal attention, RFA computes the random feature maps recurrently in autoregressive decoding.

$$\text{RFA}\left(\boldsymbol{q}_t, \{\boldsymbol{k}_i\}, \{\boldsymbol{v}_i\}\right) = \frac{\phi\left(\boldsymbol{q}_t\right)' \mathbf{S}_t}{\phi\left(\boldsymbol{q}_t\right) \cdot \mathbf{z}_t} \tag{8}$$

$$\mathbf{S}_t = \mathbf{S}_{t-1} + \phi\left(\boldsymbol{k}_t\right) \otimes \boldsymbol{v}_t, \quad \mathbf{z}_t = \mathbf{z}_{t-1} + \phi\left(\boldsymbol{k}_t\right) \tag{9}$$

Therefore, RFA needs to additionally retain $\mathbf{S}_t$ and $\mathbf{z}_t$ of each time step during the training phase, which requires extra complexity $\mathcal{O}(n)$. And LongShort calculates noncausal attention with the complexity of $\mathcal{O}(n)$ as follows:

$$P = \text{softmax}\left(KW^P\right), \bar{K} = P^\top KW^K, \bar{V} = P^\top VW^V$$
$$\text{Attention}(Q, K, V) = \text{softmax}\left[\frac{QW^Q\bar{K}^\top}{\sqrt{d_k}}\right]\bar{V} \tag{10}$$

As causal attention, LongShort needs recurrent computation. It divides the whole sequence into several chunks with chunk size $l$ and calculates $\bar{K}$ performs attention inside each chunk to satisfy autoregressive decoding.

$$\bar{K}_t = \left[P_1^\top K_{1:l}; \ldots; P_{s_t}^\top K_{(l-1)s_t:ls_t}\right]W^K, \quad \bar{V}_t = \left[P_1^\top V_{1:l}; \ldots; P_{s_t}^\top V_{(l-1)s_t:ls_t}\right]W^V \tag{11}$$

Where $\mathbf{s}_t = \lfloor t/l \rfloor$ indicates the index of the window where the $t$-th position is located. LongShort recurrently calculates $\bar{K}$ for each chunk. The overall time consumption is $\mathcal{O}(n^2/l)$, which is quadratic and referred to as efficiency degradation here.

## B    SPEEDUP IN AUTOREGRESSIVE GENERATION

We employ simulation experiments to report the speedup in autoregressive generation, where attention mechanisms are fed a set of dummy sequences with lengths of $\{8192, 12288, 16384\}$. When inputting a sequence with 8192 tokens, each attention mechanism is required to recurrently calculate from 1 to 8192 tokens, which is used to evaluate speedup autoregressive generation. The decoding process is affected by the implementation, such as incremental decoding. The results are shown in Table 10. The complexity of S4D in the inference phase is O(n), but we do not find the implementation in official code. So we did not include S4D in the speed evaluation of autoregressive generation. The results show that LAVO is faster than most baselines except FlashAttention. When the input sequence is 16384, LAVO even surpasses FlashAttention.

Table 10: Speedup on different sequence lengths in autoregressive generation.

| Method | 8192 | 12288 | 16384 |
|---|---|---|---|
| vanilla | 1.00 | 1.00 | 1.00 |
| ABC | 1.18 | 2.29 | 3.30 |
| local | 3.35 | 6.86 | 10.34 |
| LongShort | 0.70 | 1.71 | 2.94 |
| FlashAttention | **7.94** | **13.10** | 15.01 |
| FlashAttention+RoPE | 5.73 | 10.28 | 12.66 |
| LAVO | 3.71 | 10.80 | **20.81** |

## C  RESULTS ON LONG-RANGE ARENA

We set $r = 32$ and $w = 256$ for LAVO on all tasks. The results are shown in Table 11. We find that LAVO outperforms other baselines on Text, ListOps, and Pathfinder and has the best average scores.

Table 11: Results on long-range arena.

| Method | Text | ListOps | Retrieval | Pathfinder | Image | AVG |
|---|---|---|---|---|---|---|
| vanilla | 61.95 | 38.37 | 80.69 | 65.26 | 40.58 | 57.37 |
| Kernelized Attention | 60.22 | 38.78 | 81.77 | 70.73 | 41.29 | 58.56 |
| Nystromformer | 64.83 | 38.51 | 80.52 | 69.48 | 41.30 | 58.93 |
| Linformer | 58.93 | 37.45 | 78.19 | 60.93 | 37.96 | 54.69 |
| ProbSparse | 62.64 | 32.53 | 77.57 | 57.83 | 38.10 | 53.73 |
| Performer | 64.19 | 38.02 | 80.04 | 66.30 | 41.43 | 58.00 |
| Reformer | 62.93 | 37.68 | 78.99 | 66.49 | **48.87** | 58.99 |
| BigBird | 63.86 | 39.25 | 80.28 | 68.72 | 43.16 | 59.05 |
| Skyformer | 64.70 | 38.69 | **82.06** | 70.73 | 40.77 | 59.39 |
| LAVO | **65.28** | **40.57** | 80.17 | **72.14** | 40.55 | **59.74** |

