# OpenReview forum: "Linear Attention via Orthogonal Memory"
_ICLR.cc/2024/Conference — Submitted to ICLR 2024_

### Official Review · Reviewer_MBtz · 2023-10-14

**Soundness:** 1 poor
**Presentation:** 3 good
**Contribution:** 2 fair
**Rating:** 3
**Confidence:** 4

**Summary:**

The authors proposed a novel method for performing an efficient attention mechanism for autoregressive modeling utilizing orthogonal decomposition, namely LAVO. Based on experiments, LAVO outperformed chosen baselines on language modeling and LRA tasks.

**Strengths:**

I found the paper to be well-written and well-motivated. Experiments are mostly well-designed except for the flaws described below

**Weaknesses:**

- The baseline list is very limited. Authors ignore the existence of more strong baselines such as S5 [1] and Hyena [2]. While the original implementation of Hyena does not imply performing recurrent inference, it is clear that it could be combined with S5 to achieve a better performance [3]. As I see, S5 strongly outperforms LAVO with LRA benchmarks, which makes me hypothesize that it will also outperform the proposed method with language modeling with long sequences.
- The reproducibility of this paper is poor since no supplementary material with source code was released.

At the current point, I vote for the rejection of this paper since omitted baselines are vital for me.

[1] https://arxiv.org/pdf/2208.04933.pdf

[2] https://arxiv.org/pdf/2302.10866.pdf

[3] https://github.com/lindermanlab/S5/tree/development#wikitext-103

**Questions:**

Please refer to the weaknesses section

---

### Official Review · Reviewer_o7Yu · 2023-10-31

**Soundness:** 3 good
**Presentation:** 3 good
**Contribution:** 3 good
**Rating:** 5
**Confidence:** 3

**Summary:**

The paper proposes a novel linear attention mechanism called Linear Attention Via Orthogonal Memory (LAVO) to address the efficiency degradation problem in causal language modeling. LAVO uses orthogonal decomposition to compress the context into a fixed-size orthogonal memory, effectively minimizing redundancy within the context. The model also incorporates context dissecting and embedded position encoding to improve its performance. Experiments show that LAVO outperforms other efficient baselines in language modeling, text-to-speech, summarization, point cloud completion, and time-series forecasting tasks.

**Strengths:**

- LAVO achieves strong performance while maintaining linear complexity, making it suitable for long-range language modeling tasks.
- The orthogonal memory compression technique effectively reduces redundancy in the context, leading to improved efficiency.
- The context dissecting technique allows LAVO to incorporate fine-grained local context information, enhancing its encoding capabilities.
- The embedded position encoding improves the extrapolation ability of LAVO, making it more versatile and adaptable to real-world scenarios.
- LAVO can be easily extended to cross-attention tasks, demonstrating its versatility in various applications.

**Weaknesses:**

- LAVO may still lag behind FlashAttention when processing relatively short texts, limiting its efficiency in short-sequence tasks.
- The orthogonal memory compression technique may not be suitable for all types of data or tasks, as it relies on the assumption that orthogonal bases can effectively capture the context information.
- The context dissecting technique may introduce additional computational overhead, depending on the window size and the specific task.
- The embedded position encoding may not always be necessary or beneficial for all tasks, as it depends on the importance of relative positional relationships in the given context.
- The paper does not provide a comprehensive analysis of the trade-offs between different hyperparameters and their impact on the overall performance of LAVO.
- The proposed LAVO may not work well in terms of model scaling ability.

**Questions:**

NA

---

### Official Review · Reviewer_uFEK · 2023-10-31

**Soundness:** 4 excellent
**Presentation:** 4 excellent
**Contribution:** 2 fair
**Rating:** 5
**Confidence:** 5

**Summary:**

This work follows a long line of works towards efficient modeling of long sequences. Given a sequence $X$ of $n$ vectors of dimension $d$, authors propose to form an aggregate representation of context $X[<=t]$ for the vector $X_t$ by linearly projecting down $X[<=t]$ to $r$ dimensions and mean-pooling the representions to get a size-$r$ history $H_t$. Vector $H_t$ is incorporated into $X_t$ via attention (though there might be other ways of doing this e.g. concatenate + FF). In line with standard practice, authors also use local attention.

Authors compare the method to dense attention (and other baselines) on LM, text summarization, speech generation, point-cloud completion, forecasting tasks over a wide range of sequence lengths, and report modest gains on some of them and good 0-shot extrapolation to longer inputs.

**Strengths:**

1. compared to other relatively involved aggregation methods such as state spaces, the authors propose to use cummulative mean which compared to state space methods has a simpler and faster implementation based on cumsum.

2. Experimental evaluation is comprehensive and honest, encompassing tasks over various modalities over a wide range of sequence lengths. Wherever possible authors evaluate 0-shot extrapolation to longer inputs and show that their method extrapolates better than the baselines (due to locality of attention and cummulative itself).

**Weaknesses:**

1. On sequences of length upto 8k, the proposed method lags behind hardware-optimized attention in terms of both performance and throughput. For most practical applications this regime is highly relevant.

2. cummulative sum/mean is a special case of state spaces [S4, DSS, S4D] (similarly exponential moving average MEGA [Ma et al ICLR 2023]). Several follow-up works also use local attention layers similar to what authors use. Hence the paper doesnt offer a new technqiue but as pointed out above it does study a special case of using only cumsum/cummean.

3. Performance compared to learnt global convolutions (state spaces, etc) is significantly lower on Long Range Arena pointing to the importance of decayed convolutions as compared to only fixed convolutions such as cummulative sum/mean without a decay factor. This is also evident by the author's need to additionally use positions enbeddings which models based on state spaces dont require (e.g. GSS [Mehta et al ICLR 2023]). State space + local attention has been conistently shown to extrapolate to much longer inputs (e.g. GSS: PG19, train length 4096, eval length 65k) where the performance improves with longer inputs as the model is able to leverage more context.

4. Positional embeddings - the authors use additive relative bias in attention (c.f. T5, Alibi) which is undesirable due to IO aspects and will significantly slow down FLASH implementation. Rotary embeddings instead induce multiplicative relative bias and are friendly to FLASH (one can chose not to use FLASH and use vanilla attention but there is no reason why other users will not leverage it.)

**Questions:**

Typo Section 3.2 : complexity is O(wn + rn*logn) during training as parallel prefix sum (cumsum) is nlogn. I do understand that log will probably not show up in runtime as other things will dominate.

1. Please discuss the implementation of cummean. e.g. whether you're using FFT or else if you're using cumsum discuss how you ensure that the sums dont do outside the fp16 bounds (pre-normalize by the max, etc). e.g. for fp16 the largest possible value is 65k and so its not possible to post-divide by the index for input length >66k. If you used bfloat16 instead please mention.

---

### Official Review · Reviewer_FEXD · 2023-11-10

**Soundness:** 3 good
**Presentation:** 2 fair
**Contribution:** 3 good
**Rating:** 6
**Confidence:** 3

**Summary:**

This paper proposes Linear Attention Via Orthogonal memory (LAVO) to address the efficiency degeneration problem in causal language modeling with long context. LAVO compresses context into a fixed-size orthogonal memory to minimize redundancy within context. Relative position encoding is embedded into LAVO. LAVO can be used for both self-attention and cross-attention. LAVO outperforms other efficient baselines on a wide range of tasks, including language modeling, TTS, summarization, point cloud completion, and time-series forecasting.

**Strengths:**

LAVO shows empirically good accuracy and efficiency among linear attention models, making it suitable for unbounded language modeling. LAVO with longer context gives better accuracy, meaning good extrapolation performance. LAVO can be universally used for any task with attention.

**Weaknesses:**

A theoretical explanation about why the less expressiveness coming from compression and approximation does not degrade accuracy a lot might be helpful.

**Questions:**

LAVO does not have separate vectors for keys and values. Is it okay?

How do you expect the performance when LAVO is combined with FlashAttention?

How much is LAVAO sensitive to width w? How can we decide the best w? Can it be adaptive for different contexts based on the amount of information?

If there is no important detail in Table 1, it can be moved to Appendix.

There are many experiments (Tables on page 7) with causal self patterns and non-causal self patterns. Could you elaborate more on which tasks are used with causal and non-causal self patterns and why?

---

### Meta-Review · Area_Chair_RRRM · 2023-12-14

**Metareview:**

This paper proposes a new linear attention method using an orthogonal decomposition technique. Reviewers pointed out as the main weaknesses the undewhelming performance of the method and the lack of strong baselines. Unfortunately the authors do not respond to the reviewers, so these concerns remain. Therefore I recommend rejection.

**Justification For Why Not Higher Score:**

Weak baselines; no response from the authors.

**Justification For Why Not Lower Score:**

N/A

---

### Decision · Program_Chairs · 2024-01-16

Reject